# Comparative Study of Cis- and Trans-Priming Effect of PEG and BABA in Cowpea Seedlings on Exposure to PEG-Induced Osmotic Stress

K. P. Raj Aswathi, Akhila Sen and Jos T. Puthur *

Plant Physiology and Biochemistry Division, Department of Botany, University of Calicut, C.U. Campus P.O., Malappuram 673635, Kerala, India
* Correspondence: jtputhur@uoc.ac.in; Tel.: +91-944-7507845; Fax: +91-494-2400269

**Abstract:** The growth and performance of cowpea seedlings are negatively impacted by climate change and the subsequent occurrence of drought stress. Osmotic stress leads to the formation of reactive oxygen species, causing membrane breakdown, and impairs metabolic activities. The harmful effects of osmotic stress can be reduced by using seed priming techniques. Seeds of cowpea var. Anaswara were treated with polyethylene glycol (PEG) and β-amino butyric acid (BABA) as priming agents. The seedlings emerged from the primed seeds have been found to reduce the lipid peroxidation rates and improve plant water status by accumulating osmolytes such as proline, total free aminoacids, and total soluble sugars, and also enhanced the production of non-enzymatic antioxidants such as total phenolics, ascorbate, and glutathione, as well as increased the activities of enzymatic antioxidants such as catalase, peroxidase, and superoxide dismutase, which effectively scavenge ROS and maintain the homeostasis of the cell. PEG priming (cis-priming) and BABA priming (trans-priming) exhibited differential physiochemical responses in cowpea subjected to PEG stress. The current work investigates the extent of stress tolerance acquired through seed priming, and it will help to make a sensitive variety to a more tolerant one. Physiochemical responses of seedlings emerged from BABA-primed seeds towards PEG stress were better regulated to encounter the PEG-induced osmotic stress than the seedlings emerged from PEG-primed seeds.

**Keywords:** BABA priming; cross stress tolerance; PEG priming; photosynthesis; antioxidant machinery

## 1. Introduction

Plant growth is adversely affected by frequent exposure to a wide range of abiotic stressors. Moreover, plant development and metabolism are reprogrammed in response to adverse environmental conditions. Osmotic stressors, especially drought stress, significantly impair the uptake of water, seed germination, seedling growth, photosynthetic activity, and yield [1,2]. Inefficient water use efficiency, increased production of reactive oxygen species (ROS) such as superoxide and hydrogen peroxide, membrane damage and resultant increase in malondialdehyde content (a measure of lipid peroxidation), reduced photosynthetic rate, etc., are the major impact of osmotic stress on plant growth. Various stress-alleviating responses are shown by plants, including the maintenance of osmotic balance by promoting the production of osmolytes (viz. proline, aminoacids, sugars) and enhancing the scavenging of ROS through the improved activity of enzymatic and non-enzymatic antioxidants [3,4]

*Vigna unguiculata* (L.) Walp. commonly called cowpea, a protein rich legume crop, is a promising legume contributing for alleviating hidden hunger in the form of protein deficiency [5]. Although cowpea is a warm weather, semi-arid crop, the growth of cowpea is highly dependent on changes in water availability. Moreover, osmotic stress limits the growth and yield of cowpea [6]. Therefore, it is necessary to find ways to improve the stress tolerance potential of this crop. One of the prominent techniques employed to develop

osmotic stress tolerance in plants is seed priming. The treatment of seeds with natural and synthetic priming agents boosts the stress tolerance potential of plants via modulating their physiochemical characteristics [7]. The selection of the apt priming method becomes important to achieve the improvement of desired qualities in the target plant.

Seed priming treatments enhance the rate of seed germination, seedling growth, and performance even under stressful conditions [8]. The enhancement in the production of antioxidants, osmolytes, and effective maintenance of water status serves as a good strategy to overcome osmotic stress [9]. Various seed priming techniques such as hydro priming, osmo priming, and chemical priming are being practiced to make plants tolerant to osmotic stress. Osmo priming involves the soaking of seeds in osmotic solutions, enabling controlled hydration. Studies have revealed the positive effect of osmo priming in bringing better germination patterns, improved seedling growth, and enhanced stress tolerance [10]. The osmo priming treatment itself acts as a mild stressor and boosts the stress memory in plants and can thereby trigger the tolerance mechanism on exposure to further stress. Polyethylene glycol (PEG) is an effective priming agent, which can impose mild osmotic stress in plants and thereby elicit stress tolerance against further exposure to a stress [11]. Likewise, several chemical agents also have the potential role to serve as priming agent [12]. β-amino butyric acid (BABA) is a non-protein amino acid, which has a specific role in providing augmented plant immunity to counteract the adverse effects of various types of stresses [13]. BABA priming prepares the plant to defend against a variety of stressors. The cis-priming effect of PEG against PEG-induced osmotic stress reveals the positive response of prior exposure to mild PEG stress so as to overcome the impact of the same stress at a later period. However, BABA priming involves trans-priming so that the priming stimuli and the preceding stress are different in nature, yet they could develop cross-tolerance towards stress response in cowpea seedlings. The ability of BABA and PEG to act as a priming agent in a variety of plant species (viz. *Arabidopsis thaliana*; *Malus pumila*; *Vicia faba*; *Zea mays*) against diverse abiotic stresses such as salinity and drought have already been reported [14–17]. However, the present work is the first report on the beneficial role of these priming agents in imparting tolerance to cowpea seedlings against PEG-induced drought stress. These findings not only highlight the priming effect of BABA and PEG towards drought tolerance in cowpea, but also give a comparative account on the cis- and trans-priming effects. In this work, our research team sheds light on the osmotic stress alleviating role of BABA and PEG priming in cowpea. Additionally, it gives an understanding of the BABA priming-induced cross-stress tolerance mechanism towards PEG-induced osmotic stress.

## 2. Materials and Methods

### 2.1. Seed Priming

The seeds of cowpea (*Vigna unguiculata* (L.) Walp. 'Anaswara') were collected from Kerala Agricultural University, Thrissur, Kerala, India. Surface sterilization of the seeds was performed before seed priming. Ten healthy seeds were randomly selected and sterilized in 0.1% $HgCl_2$ solution for 1 min, followed by thorough washing in distilled water. For osmo priming, the sterilized seeds were immersed in polyethylene glycol solution (PEG 6000) of different concentrations (5%, 10%, 15%, 20%, and 25% with osmotic potential −0.2 MPa, −0.45 MPa, −0.675 Mpa, −0.9 MPa, and −1.12 MPa, respectively) for 6 h. For β-amino butyric acid (BABA) priming, the seeds were soaked in different concentrations of BABA (0.5 mM, 1 mM, 1.5 mM, 2 mM and 2.5 mM) for 6 h.

### 2.2. Experimental Design

Unprimed seeds and the seeds treated with 10% PEG and 1.5 mM BABA (optimum concentrations for priming) were grown in separate polypropylene culture bottles (22 × 12 cm). The absorbent cotton in the bottles contains unprimed seeds and BABA and PEG primed seeds soaked in distilled water (control) and 15% PEG 6000 (stress). The bottles were kept in a light intensity of 300 µmol m$^{-2}$ s$^{-1}$ at 25 ± 2 °C and 55 ± 5% relative humidity for 14/10 h

of light–dark cycles in a plant growth chamber (INLABCO, India). The morphological and biochemical analyses were carried out on the 8 d old seedlings.

### 2.3. Estimation of Chlorophyll Content

Total chlorophyll content was estimated according to Arnon [18]. An amount of 0.05 g tissue was homogenized in 5 mL of 80% acetone and centrifuged at 8496 g. The absorbance of the supernatant was measured using a spectrophotometer (Multiskan, Thermo Fisher Scientific, Vantaa, Finland).

### 2.4. Estimation of Malondialdehyde Content and Reactive Oxygen Species

Malondialdehyde content (MDA) was calculated according to the standard protocol of Heath and Packer [19]. An amount of 0.5 g tissue was homogenized in 5 mL of 5% trichloro acetic acid (TCA) and the homogenate was centrifuged at 20,392 g for 15 min. A volume of 2 mL of the supernatant was mixed with the same amount of 0.5% of thiobarbituric acid (TBA) in 20% TCA. This mixture was heated for 24 min at 95 °C, allowed to cool, and then centrifuged for 2 min at 5098 g. The absorbance of the supernatant was noted, and the MDA content was calculated using its molar extinction coefficient of 155 mM$^{-1}$ cm$^{-1}$.

Superoxide content ($O_2{}^-$) was determined according to Doke [20]. The tissue was cut into small pieces and immersed in a solution of 0.01 M potassium phosphate buffer (pH 7.8), containing 0.05% NBT and 10 mM sodium azide. The absorbance was measured at 580 nm after incubation.

Hydrogen peroxide ($H_2O_2$) was assessed as per Junglee et al. [21]. Samples were homogenized in 5 mL of 0.1% ice-cold TCA and centrifuged at 20,392 g for 15 min. The supernatant was collected and used for the estimation of $H_2O_2$.

### 2.5. Determination of Leaf Osmolality

Based on the procedure of Hura et al. [22], leaf osmolality was measured using a vapor pressure osmometer (Wescor 5520, Logan, UT, USA). The cell sap was extracted using a freeze-thawing procedure. A 10 μL pipette was used to collect sap extruded from the sample, and it was promptly transferred to the disc chamber of osmometer where readings were taken.

### 2.6. Estimation of Non-Enzymatic Antioxidants

Proline content was calculated using the protocol of Bates et al. [23] with L-proline as the standard. The plant sample was homogenized in 5 mL of sulfosalicylic acid and the supernatant was collected after centrifugation at 16,993 g for 10 min at 4 °C and used for the estimation.

The total soluble sugars content was determined based on the procedure of Dubois et al. [24] using D-glucose as the standard. The protocol of Moore and Stein [25] was used to estimate free amino acids, and L-Leucine was used to create the standard curve. The extraction of total soluble sugars and total free aminoacids was carried out in 80% ethanol and the homogenate was centrifuged at 16,993 g for 10 min at 4 °C. For the estimation of total soluble sugars, a known volume of the aliquot was taken and made up to 1 mL. To this, 5% phenol and concentrated sulphuric acid was added for the estimation and the optical density was noted at 490 nm. The estimation of aminoacids was performed with ninhydrin reagent and the absorbance was measured at 570 nm.

The amount of ascorbate and glutathione was calculated according to Chen and Wang [26], and L-ascorbic acid, as well as reduced glutathione, respectively, were used as the standards. A known volume of aliquot was used for the estimation of reduced ascorbate content and was mixed with sodium phosphate buffer. To this mixture, an equal volume of 10% (*v/v*) TCA, 42% (*v/v*) $H_3PO_4$, and 4% (*w/v*) bipyridyl were added. The mixture was incubated for 15 min and the absorbance was noted at 524 nm. Reduced glutathione content was estimated with sodium phosphate buffer (pH 6.8) and 5,5′-dithiobis (2-nitrobenzoic acid, DTNB). The absorbance at 412 nm was noted for calculating the glutathione content.

Total phenolics content was calculated as per Folin and Denis [27], with catechol as the standard. The extraction was performed with 80% alcohol and the homogenate was centrifuged at 16,993 g for 10 min at 4 °C. The supernatant was collected and used for the estimation. Folin and Ciocalteu reagent (1 N) and 20% sodium carbonate were used for the estimation, and the color of the resultant mixture was read at 650 nm.

### 2.7. Determination of Enzymatic Antioxidants

The crude enzyme extract was made from seedlings using the procedure of Yin et al. [28]. The method of Bradford [29] was used to estimate the protein in the enzyme extract, with BSA as the standard. Ascorbate peroxidase (APX) (EC 1.11.1.11) activity was measured according to Nakano and Asada [30], and the amount of enzyme needed to oxidize 1 µmol of ascorbate in 1 min was used to define 1 unit of enzyme activity.

The activity of superoxide dismutase (SOD) (EC 1.15.1.1) was measured using the protocol of Giannopolitis and Ries [31]. Superoxide dismutase's ability to prevent the photochemical reduction of nitroblue tetrazolium (NBT) was used to estimate the activity. The enzyme needed to inhibit the photochemical reduction of NBT by 50% was determined to be one unit of SOD activity.

The catalase (CAT) (EC 1.11.1.6) activity in fresh samples was determined using the method of Kar and Mishra [32]. The enzyme activity was measured in micromoles of $H_2O_2$ decomposed per minute per mg of protein.

### 2.8. Measurements of Chlorophyll a Fluorescence Parameters

Plant Efficiency Analyzer (Handy PEA, Hansatech), a portable fluorometer, was used to examine the photosynthetic performance of plants by measuring chlorophyll *a* fluorescence-related parameters, and the data were evaluated using Biolyzer HP3 software (Bioenergetics Laboratory, University of Geneva, Switzerland) [33]. The measurements were taken using the leaf clips provided by the manufacturer, which were clipped on the upper surfaces of the leaves for a 20 min dark adaption period. A 1 s pulse of white light (3000 µmol photon $m^{-2}$ $s^{-1}$) was used to cause maximum fluorescence, with the gain adjusted to 0.7 to avoid over scaling errors. Biolyzer HP 3 software (Bioenergetics Laboratory, University of Geneva, Switzerland) was used to create energy pipeline leaf models from the data. Chlorphyll *a* fluorescence parameters such as absorption flux per cross section (ABS/CSm), trapped energy per cross section (TRo/CSm) electron transport flux per cross section (ETo/CSm), and dissipated energy flux per cross section (DIo/CSm) were measured in relative units and from the data, phenomenological energy pipeline leaf models were deduced.

### 2.9. Statistical Analysis

The results were statistically analyzed using Duncan's multiple range tests at a 5% probability level. Using SPSS software 21.0, one-way ANOVA was performed. The data represent the mean and standard error.

## 3. Results

### 3.1. Selection of Stress Imparting Concentration of PEG and Standardisation of Effective Concentrations of Priming Agents

The concentration of PEG, which imparts ~50% retardation in various growth parameters and chlorophyll content, was chosen as the stress imparting concentration. Cowpea variety Anaswara exhibited 50% growth retardation at 15% PEG (Supplementary Table S1). For the priming treatments, different concentrations of BABA (0, 0.5, 1.0, 1.5, 2.0, and 2.5 mM) and PEG-6000 (0, 5, 10, 15, 20, and 25%) were experimented and the treatments were carried out for three different time periods (3, 6, and 9 h). The effective priming concentration of PEG and BABA, which brings maximum enhancement in growth attributes and chlorophyll content, was selected as the best priming concentrations and it was 1.5 mM BABA and 10% PEG treatments for 6 h (Supplementary Table S2).

### 3.2. Oxidative Stress Markers

The reactive oxygen species such as superoxide and hydrogen peroxide content were increased in leaves of seedlings exposed to PEG stress. A significant increase in superoxide (110%) and hydrogen peroxide (90%) content was observed in seedlings when compared with the control. BABA priming significantly reduced the superoxide (62%) and hydrogen peroxide (33%) generation in cowpea under PEG stress conditions (Figure 1a,b).

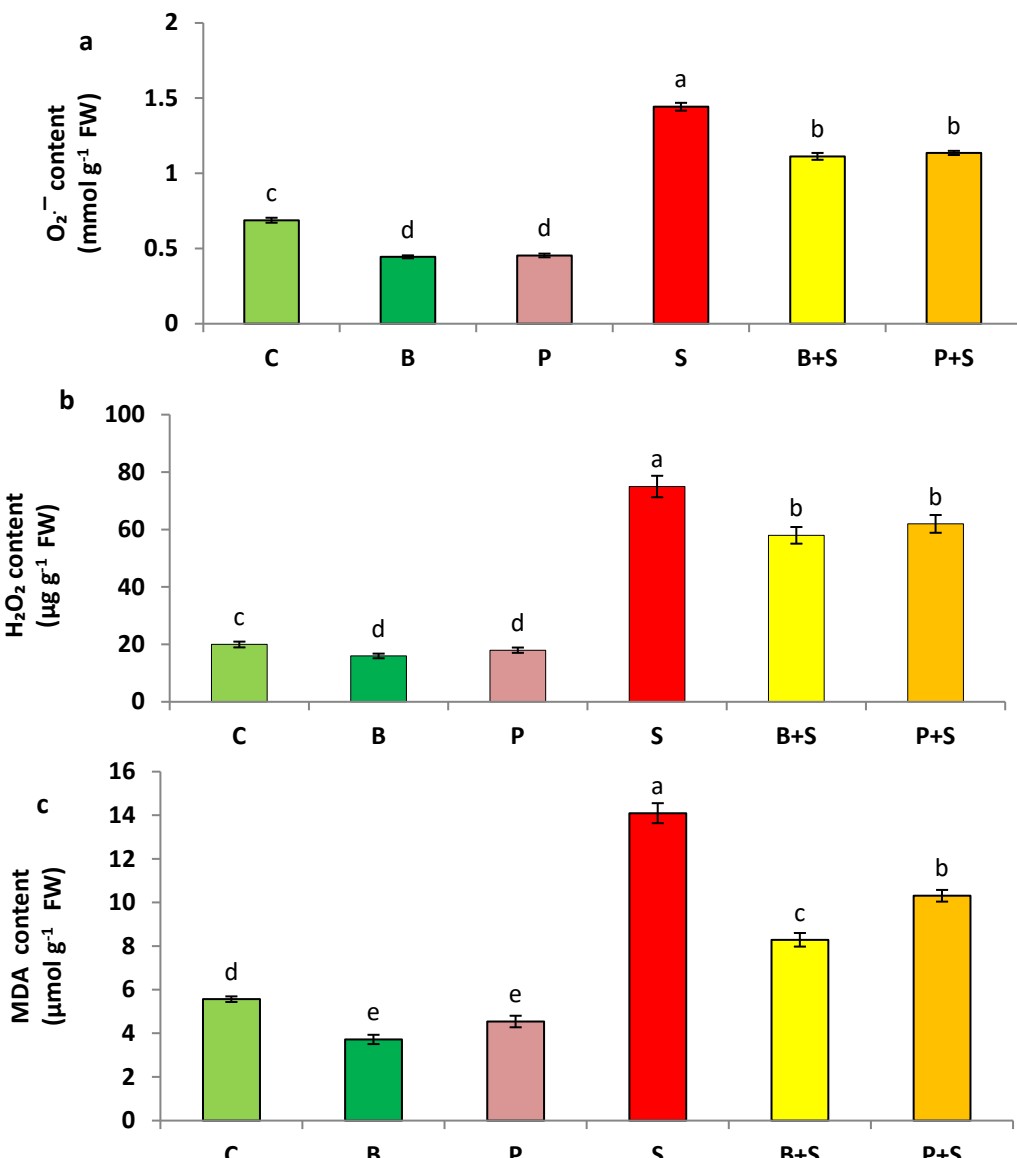

**Figure 1.** Effect of BABA and PEG seed priming on superoxide (**a**), hydrogen peroxide content (**b**), and lipid peroxidation (**c**) in cowpea seedlings exposed to PEG stress. C (control), B (BABA-primed seedlings without any stress), P (PEG-primed seedlings without any stress), S (unprimed seedlings subjected to PEG stress), B + S (BABA-primed seedlings subjected to PEG stress), P + S (PEG-primed seedlings subjected to PEG stress). Values represent the mean ± SE of three independent experiments. Alphabetical letters show significant difference between treatments performed using Duncan's multiple range test.

The MDA content in cowpea seedlings was significantly increased in response to PEG stress (153%) when compared to the respective control. However, the increment in the MDA content due to PEG stress conditions was highly regulated by the BABA and PEG primings. The seedlings that emerged from BABA-primed seeds had a significantly lower

rate of lipid peroxidation, and the increase of MDA content was only 49% with respect to the control. PEG priming also reduced the accumulation rate of MDA content, and the increase was only 85% under PEG stress (Figure 1c).

*3.3. Osmolality*

PEG stress resulted in an increase in leaf osmolality (an increase of 90 mmol kg$^{-1}$) in cowpea seedlings when compared to the control. Under PEG stress, priming with BABA significantly increased leaf osmolality by 133 mmol kg$^{-1}$ and with PEG priming, the increase was 136 mmol kg$^{-1}$ over the control (Table 1).

**Table 1.** Effect of BABA and PEG seed priming on osmolality in cowpea seedlings exposed to PEG stress. C (control), B (BABA-primed seedlings without any stress), P (PEG-primed seedlings without any stress), S (unprimed seedlings subjected to PEG stress), B + S (BABA-primed seedlings subjected to PEG stress), P + S (PEG-primed seedlings subjected to PEG stress). Values represent the mean ± SE of three independent experiments. Alphabetical letters show significant difference between treatments performed using Duncan's multiple range test.

| Treatments | Osmolality (mmol kg$^{-1}$) |
|:---:|:---:|
| C | 292 ± 7.05 |
| B | 335 ± 9.01 |
| P | 338 ± 10.40 |
| S | 382 ± 9.17 |
| B + S | 425 ± 7.64 |
| P + S | 428 ± 8.66 |

*3.4. Non-Enzymatic Antioxidants*

The proline content of cowpea seedlings increased significantly in response to PEG stress (91%) when compared to the control. Priming with BABA had an increasing effect on proline accumulation in cowpea seedlings. In comparison to the control, in seedlings emerged from BABA-primed seeds, the proline content increased by 38% under control conditions and 123% under PEG stress. In the case of PEG priming, the increment was only 24% and 112%, respectively, under control and stress conditions (Figure 2a).

When cowpea seedlings were subjected to PEG stress, the total soluble sugars content increased significantly (132%) when compared to the respective controls. The total soluble sugars content of BABA-primed seedlings increased by 182% and an increase of 181% was noted in the case of PEG priming when compared to the control on PEG stress exposure (Figure 2b).

Similarly, the total free amino acids content of cowpea seedlings increased by 100% as a result of PEG stress. BABA priming increased the total free amino acids content by 53% without stress and 134% with PEG stress. PEG priming increased total free amino acids content by 43% and 126% under normal and PEG stress conditions, respectively (Figure 2c).

An increase in ascorbate content (112%) in response to PEG stress was recorded in cowpea seedlings. Seedlings raised from seeds subjected to BABA priming and PEG priming and exposed to PEG stress still had a higher content of ascorbate. BABA priming enhanced the production of ascorbate by 59% in the control and 144% in seedlings exposed to PEG stress. Similarly, PEG priming also increased the production of ascorbate, but the increase was only 52% and 138% under non-stressed and stressed conditions (Figure 3a).

PEG stress increased the content of glutathione in cowpea seedlings by 78% when compared to the control. BABA and PEG priming had a greater impact on ascorbate accumulation in cowpea seedlings; it increased by 101% and 86%, respectively, compared to the control under PEG stress (Figure 3b). The highest increment was noted in BABA-primed seedlings. The total phenolics content of cowpea seedlings also increased in BABA-primed seedlings (28% in control and 109% under stress conditions) compared to PEG-primed seedlings (16% and 97%, respectively, in control and stress conditions) (Figure 3c).

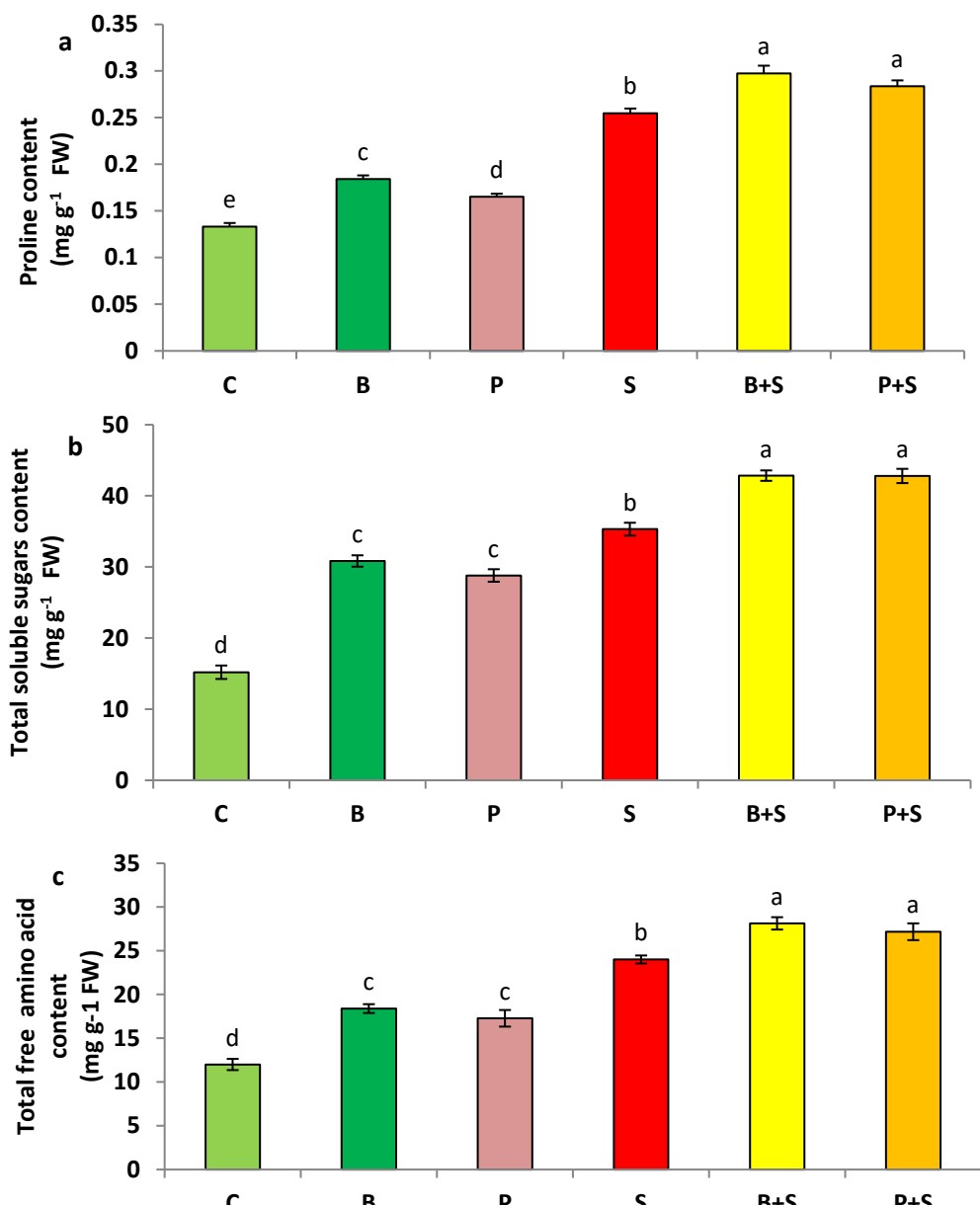

**Figure 2.** Effect of BABA and PEG seed priming on proline (**a**), total sugar content (**b**), and total amino acids (**c**) in cowpea seedlings exposed to PEG stress. C (control), B (BABA-primed seedlings without any stress), P (PEG-primed seedlings without any stress), S (unprimed seedlings subjected to PEG stress), B + S (BABA-primed seedlings subjected to PEG stress), P + S (PEG-primed seedlings subjected to PEG stress). Values represent the mean ± SE of three independent experiments. Alphabetical letters show significant difference between treatments performed using Duncan's multiple range test.

*3.5. Enzymatic Antioxidants*

SOD activity increased by 248% in cowpea seedlings subjected to PEG stress compared to controls. SOD activity significantly increased by 386% in BABA priming and 329% in PEG priming under PEG stress when compared to the control (Figure 4a).

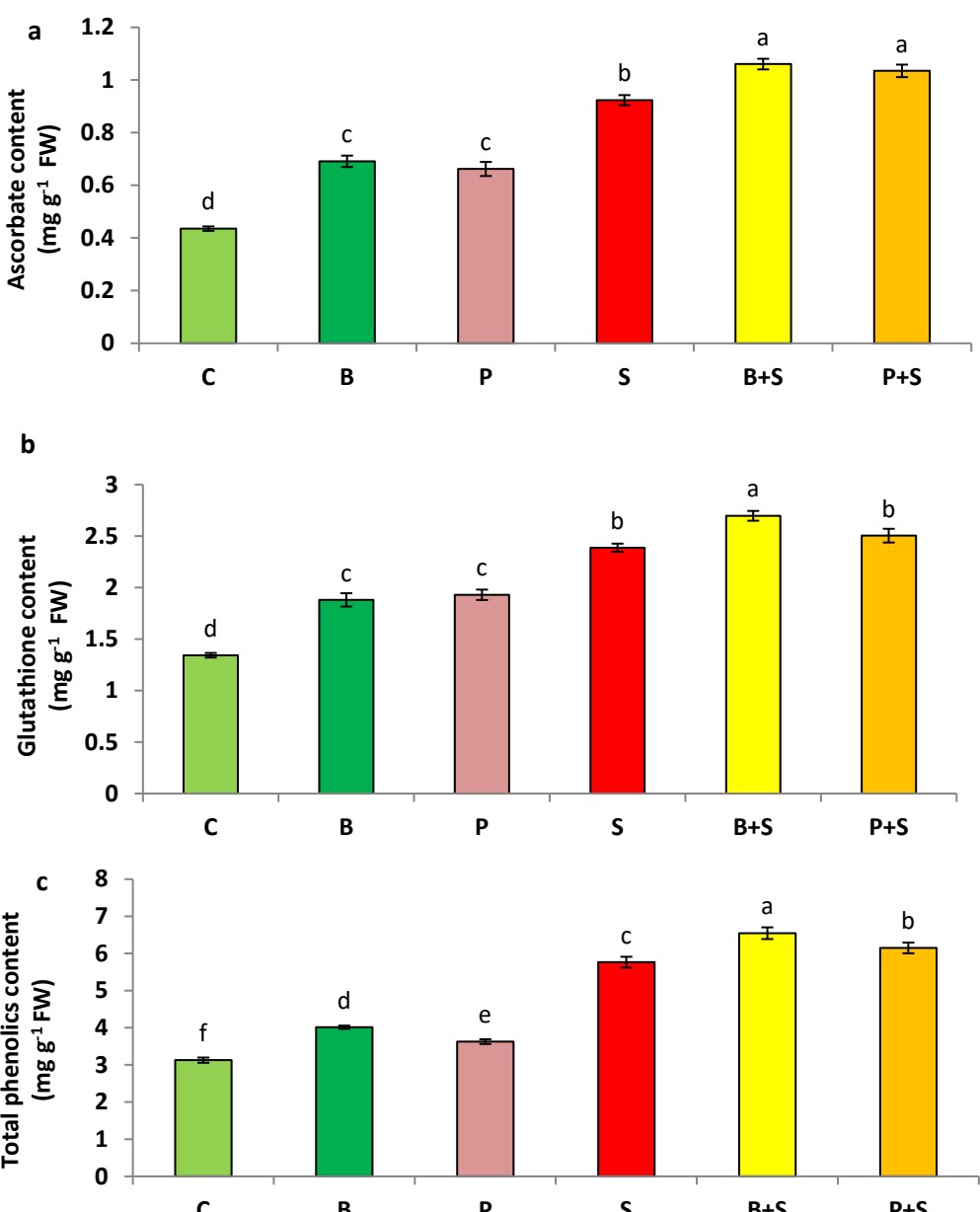

**Figure 3.** Effect of BABA and PEG seed priming on ascorbate (**a**), glutathione content (**b**), and total phenolics content (**c**) in cowpea seedlings exposed to PEG stress. C (control), B (BABA-primed seedlings without any stress), P (PEG-primed seedlings without any stress), S (unprimed seedlings subjected to PEG stress), B + S (BABA-primed seedlings subjected to PEG stress), P + S (PEG-primed seedlings subjected to PEG stress). Values represent the mean ± SE of three independent experiments. Alphabetical letters show significant difference between treatments performed using Duncan's multiple range test.

On exposure of the cowpea seedlings to PEG stress, CAT activity increased by 325% compared to the control. CAT activity was higher in BABA treatment and it increased by 450% in seedlings grown from BABA-primed seeds and was followed by a 375% increase in PEG-primed seeds when compared to the respective controls on PEG stress exposure. Similarly, when cowpea seedlings were subjected to PEG stress conditions, APX activity increased significantly (295%) when compared to the control. Both primings significantly increased APX activity, with the greatest increase observed in BABA priming (400%), followed by PEG priming (374%) (Figure 4b,c).

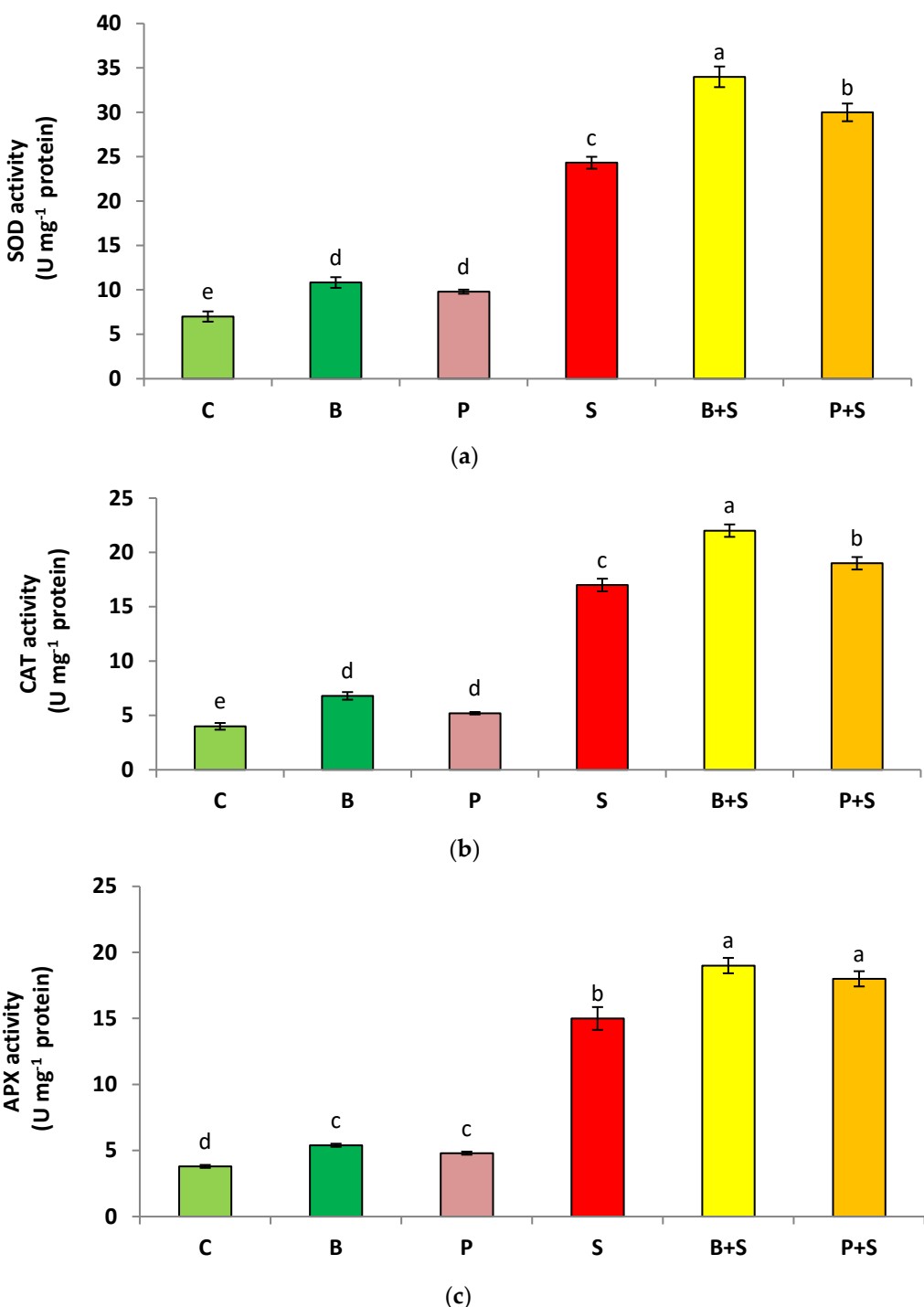

**Figure 4.** Effect of BABA and PEG seed priming on SOD (**a**), CAT (**b**), and APX (**c**) activities in cowpea seedlings exposed to PEG stress. C (control), B (BABA-primed seedlings without any stress), P (PEG-primed seedlings without any stress), S (non-primed seedlings subjected to PEG stress), B + S (BABA-primed seedlings subjected to PEG stress), P + S (PEG-primed seedlings subjected to PEG stress). Values represent the mean ± SE of three independent experiments. Alphabetical letters show significant difference between treatments performed using Duncan's multiple range test.

*3.6. Photobiology*

To analyze the effects of BABA and PEG primings on the primary photochemistry of PSII in cowpea seedlings, different parameters such as absorption flux (ABS), electron transport flux (ET), trapping flux (TR), and dissipated energy as heat or fluorescence

(DI) per PSII reaction center (RC), as well as the specific membrane model, were studied (Figure 4). It was observed that RC, ABS, TR, and ET decreased in the unprimed cowpea seedlings under PEG stress conditions when compared with the control. However, the maximal dissipation energy (DIo/RC) increased in the cowpea seedlings emerged from unprimed seeds compared to those emerged from BABA- and PEG-primed seeds. BABA priming and PEG priming resulted in a prominent increase of RC, ABS, TR, and ET, and decreased the dissipation of energy (DI) from reaction centers and even tended to be equal to their respective control cowpea seedlings on exposure to PEG stress (Figure 5). However, the photosynthetic efficiency is better in seedlings primed with BABA (trans-priming) than the seedlings primed with PEG (cis-priming). Absorption flux per cross section (ABS/CSm), trapped energy per cross section (TRo/CSm), and electron transport flux per cross section (ETo/CSm) were higher in BABA-primed seedlings than the PEG-primed seedlings. Dissipated energy flux per cross section (DIo/CSm) was less in BABA-primed seedlings than PEG-primed seedlings subjected to stress, and this helps the seedlings emerged from BABA-primed seeds perform better than the seedlings emerged from PEG-primed seeds.

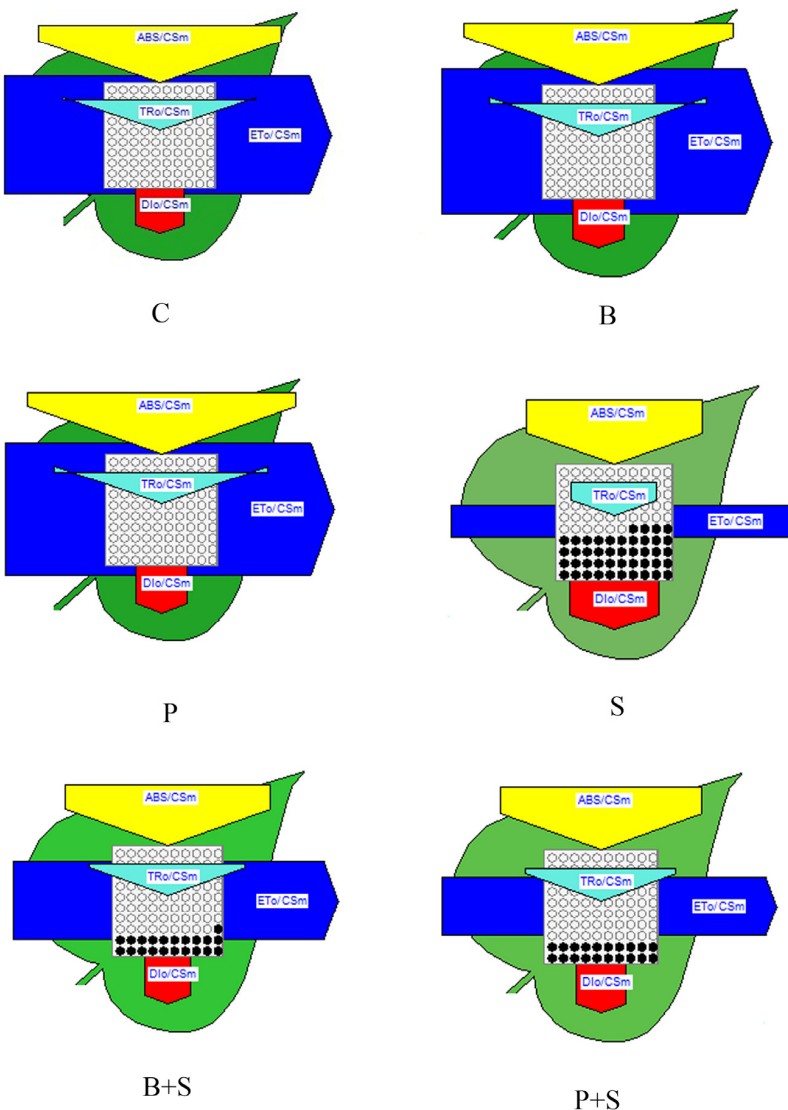

**Figure 5.** Effect of BABA- and PEG-seed priming on photosynthetic activity in cowpea seedlings exposed to PEG stress. C (control), B (BABA-primed seedlings without any stress), P (PEG-primed seedlings without any stress), S (unprimed seedlings subjected to PEG stress), B + S (BABA-primed seedlings subjected to PEG stress), P + S (PEG-primed seedlings subjected to PEG stress).

## 4. Discussion

Pre-exposure to primary mild stress improves a plant's efficiency and performance when it is subjected to secondary stress at a later stage. The chances of encountering a second stress by plants may vary and the stress encountered may or may not be the same as the stress encountered initially. If the nature of initial and later stresses are the same, then it is called cis-priming. However, in the case of trans (cross)-priming, the priming stimuli is different from that of the later stress to which the plants are exposed, and it entails cross-talk between different stressors. The success of cross-stress priming is due to the initiation and interaction of synergistic stress signalling pathways that are shared between different types of stressors, and accordingly, primed plants acquire the capacity for augmented stress tolerance due to cross-stress memory [34]. The present study examines the cis-priming and trans-priming effect of PEG and BABA on seedlings of cowpea variety Anaswara subjected to PEG stress.

Several studies in various crops show the beneficial role of cis- and trans-priming against various abiotic stresses. Halopriming with NaCl enhanced NaCl stress tolerance in tolerant and sensitive varieties of *Vigna radiata* [35]. A similar kind of cis-priming using UV-B in the case of *O. sativa* ameliorated the damages induced by UV-B stress in plants [36]. The stress-alleviating potential of trans-priming is also significant, as evidenced by the work of many researchers. Cross-tolerance to stress, attributed to the augmented activity of antioxidants and photosynthetic machinery, was reported in the case of UV-B-primed rice seedlings exposed to NaCl and PEG stresses [37]. The mild dosage of UV-B radiation triggers various physiological responses, including an enhanced defence mechanism in UV-B primed plants exposed to salinity stress compared to the plants exposed to the same UV-B irradiations, but at higher dosage, induces stress. This also demonstrates the potential of a cross-tolerance response towards different stresses encountered by plants [38].

Chemical priming techniques have greater influence on maintaining plant water status, improved photosynthetic efficiency, and an enhanced ROS scavenging mechanism, and thereby shows increased cis-/cross-stress tolerance mechanisms in plants [39]. In cotton, melatonin seed priming was found to be effective against salinity stress [40]. Similarly, seed treatment with BABA contributes towards salinity stress tolerance in barley, and it was achieved through the upregulated activity of enzymatic antioxidants [41]. Seed priming with several osmotic agents such as PEG, NaCl, potassium chloride (KCl), calcium chloride (CaCl$_2$), etc. also aided in building up tolerance towards various abiotic stresses such as drought and salinity in plants [42,43].

PEG causes blockade in water movement pathways, reducing water absorption and causes desiccation, similar to actual drought stress [44]. As a result, PEG is being frequently used in many studies as an artificial drought stress inducer [45]. The present study proved that in cowpea seedlings, the PEG-induced overproduction of ROS damaged the membrane system and as a result of it, degradation of the polyunsaturated fatty acids in membranes occurs, as evidenced from the increased rate of MDA content. The increase in MDA levels during drought indicates enhanced membrane lipid peroxidation through increased generation of ROS [46,47]. Low levels of ROS (Figure 1a,b) and MDA contents (Figure 1c) in seedlings emerged from primed seeds indicate that these seedlings are more tolerant to PEG-induced oxidative damages, whereas increased membrane damage in unprimed seedlings reveals the loss of ROS homeostasis in the cell. These results are in conformity with the findings of Sen et al. [37], wherein they reported that increased ROS and MDA rates were higher in unprimed rice seedlings of both stress sensitive and tolerant varieties on exposure to PEG stress.

The leaf osmolality of seedlings increases as a result of PEG stress, and tries to cope with PEG-induced osmoticum. Both primings could bring about an enhancement in leaf osmolality even when the seedlings were not exposed to PEG stress, indicating that priming initiates the process of osmoticum build-up. The synthesis process of necessary osmolytes gets initiated and is readied for enhanced production at any point of encountering stress. Seedlings subjected to BABA and PEG priming and exposed to PEG stress further enhanced

the osmolality of leaf, which aided in overcoming osmotic stress by increased osmolytes accumulation. Osmolytes such as proline, sugars, and amino acids act as osmo protectants, protecting plants from abiotic stresses by assisting with cellular osmotic adjustment, ROS detoxification, membrane integrity protection, and enzyme/protein stability [48]. An increased accumulation of proline, sugars, and amino acids in primed seedlings when compared to unprimed seedlings justifies the enhanced leaf osmolality in the former as a result of seed priming.

According to Apel and Hirt [49], the degree of ROS-mediated damage is determined based on the balance between the rate of ROS production and the activation of the antioxidant defence system. Increased levels of ROS and membrane damage in seedlings exposed to PEG stress significantly increased the levels of antioxidant enzymes (SOD, CAT, and APX) and non-enzymatic antioxidants (AsA and GSH) in cowpea seedlings, but still, the balance could not be maintained in favor of the antioxidation process. Drought stress causes the production of reactive oxygen species (ROS) such as superoxide anion radicals, hydrogen peroxide, and hydroxyl radicals, and plants have the ability to scavenge these reactive oxygen species by producing antioxidant enzymes such as SOD, CAT, POD, and APX [50–52]. A greater reduction of ROS and MDA accumulation in seedlings emerged from primed seeds was linked to increased antioxidant enzyme activities such as SOD, CAT, and APX, indicating priming mediated a better antioxidant defence mechanism. Less MDA accumulation in BABA-primed seedlings was associated with better antioxidant enzyme activities resulting in successful scavenging, indicating the enhanced stress tolerance potential attained as a result of priming, enabling the seedlings to grow under water-deficit conditions. Trans-priming brought about better ROS scavenging activity over cis-priming. This confirms the multiple signalling response getting activated with BABA priming, which will impart greater tolerance to cowpea encountered with stress of a different nature from that of the priming type. Researchers have discovered multiple stress responses in plants, linked to BABA and GABA treatment, including a synergistic advantage against drought and salt stress due to the elevated expression of genes associated with osmotic stress [53].

The sharper decline in the ABS in response to PEG stress represents the decreased energy absorption efficacy of PS II in cowpea seedlings during stress. PEG stress leads to the deactivation of active reaction centres in cowpea seedlings. Additionally, it affects the photochemistry of cowpea through a reduced rate of photon trapping (TR) and electron transport (ET). The inactive reaction centers do not efficiently trap photons; therefore, an increase in the level of untrapped photons occurs, which, in turn, leads to the surge of DIo/RC [54]. In the case of seedlings raised from BABA- and PEG-primed seeds, the energy absorption, photon trapping, and electron transport increased, denoting an increase in the number and/or activity of reaction centers even under PEG stress. Different types of priming techniques have the potential for this and have already been proved in the case of *Pisum sativum*, *Triticum aestivum*, etc. [55,56].

The mild tress experienced during the priming process induces a stress memory in PEG-primed seeds and imparts stress tolerance to seedlings during PEG stress exposure. The process of stress memory imprinting in seeds upon priming treatment is the prime cause of the augmented stress tolerance mechanism shown by primed plants exposed to stress at a later stage. The development of stress memory through seed priming techniques are documented from our research works, as well as of others [57,58]. BABA priming has been shown to improve NaCl and PEG stress tolerance in *V. radiata* and *O. sativa* seedlings by regulating osmotic adjustments in seedlings emerged from seeds primed with 1 mM BABA [13,59]. Likewise, in *Piper nigrum*, a priming application of 2 mM GABA was found to be effective against PEG-induced osmotic stress. Moreover, priming with non-protein amino acids such as BABA and GABA has gained popularity because it allows plants to effectively resist abiotic stresses without suffering from costly energy investments on defence mechanisms [60]. Furthermore, BABA has the potential to influence at the genomic level by inducing drought-tolerance-related genes such as HSP, LEA, NAC, and WRKY

and increasing their expression, attributing to the enhanced osmotic stress tolerance in plants [11].

The initial exposure of seeds to PEG during priming treatments brings better stress tolerance in plants emerged from primed seeds and they show enhanced stress tolerance upon a stress exposure at a later stage. Here, the initial priming stimulus evokes improved osmotic adjustments through the accumulation of osmolytes, reducing ROS levels and thereby maintaining membrane integrity, and exhibits improved photosynthetic performance and enhanced antioxidants activity, which aids in developing cis-priming memory and contributes stress tolerance towards a further encounter with the same kind of stress. However, in the case of trans-priming, the BABA acts as the elicitor factor which modify the signal transduction process when the seedlings were exposed to a further stress. It amplifies and brings about synergistic action on the stress signalling pathways, which results in cross-talk between signalling cascades, bringing about augmented stress tolerance responses in BABA-primed seedlings. Such an action of a trans-priming effect by BABA results in an improved cross-stress tolerance mechanism in cowpea seedlings more efficiently. This was accomplished by activating natural defence mechanisms against abiotic stresses, but not activating the entire defence arsenal prior to the stress exposure. BABA is a well-known signalling molecule that regulates the expression of genes involved in drought stress resistance [53]. As a result, BABA priming in cowpea seeds can certainly bring about enhanced drought stress tolerance compared to PEG priming. Understanding the seed priming technique and comparing cis- and trans-priming effects in cowpea seedlings opens a new route for understanding the mechanism behind the osmotic stress tolerance response in cowpea plants through priming. This will be helpful in formulating strategies for enhancing the osmotic stress tolerance potential of cowpea plants.

## 5. Conclusions

This research work sheds light on the cross-talk between multiple signalling pathways and provides a vivid understanding of the priming-induced cross-stress tolerance mechanism attributed through BABA and PEG priming in cowpea seeds. BABA is highly soluble in water and has been found to be directly absorbed by seeds and transported throughout the plant. Few investigations have been conducted so far on the utilisation of BABA-like neurotransmitters in cowpea. Our findings improve the knowledge of practicing the optimum seed priming technique that offers cross-stress tolerance. As a result, this work provides a foundation for enhancing the tolerance potential of a sensitive cowpea variety.

**Supplementary Materials:** The following supporting information can be downloaded at: https://www.mdpi.com/article/10.3390/seeds2010007/s1, Table S1: Standardization of stress imparting concentration of PEG; Table S2: Fixation of priming concentration of BABA and PEG based on shoot length, fresh weight, dry weight and total chlorophyll content.

**Author Contributions:** Conceptualization, K.P.R.A. and J.T.P.; investigation, K.P.R.A.; writing—original draft, K.P.R.A.; writing—review and editing, J.T.P. and A.S.; supervision, J.T.P.; funding acquisition, K.P.R.A. All authors have read and agreed to the published version of the manuscript.

**Funding:** This research was funded by University Grant Commission (UGC), India, in the form of JRF under grant 318224.

**Institutional Review Board Statement:** Not applicable.

**Informed Consent Statement:** Not applicable.

**Data Availability Statement:** The data presented in the current work are available from the corresponding author on reasonable request.

**Acknowledgments:** We express sincere thanks to the University Grant Commission (UGC), India for providing financial support in the form of the Junior Research Fellowship (JRF). Kerala Agriculture University (KAU), Thrissur, Kerala, India, is also gratefully acknowledged for providing cowpea seeds. The authors extend their sincere thanks to the Department of Science and Technology (DST),

Government of India for granting funding under the Fund for Improvement of S & T Infrastructure (FIST) programme (SR/FST/LSI-532/2012).

**Conflicts of Interest:** The authors declare no conflict of interest.

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
