# Peer review of "Comparative Study of Cis- and Trans-Priming Effect of PEG and BABA in Cowpea Seedlings on Exposure to PEG-Induced Osmotic Stress"

_2674-1024, doi:10.3390/seeds2010007_

Round 1

Reviewer 1 Report

Manuscript ID: seeds-2082877.

The authors treated cowpea seeds with either BABA or PEG as priming agents and measured parameters related to plant drought responses. The authors concluded that BABA primed seeds performed better than PEG primed seeds towards PEG stress.

One big issue with the way the authors present their findings is that, they wanted to compare between BABA- and PEG-treated seeds, however, in all of their comparisons, they were trying to compare all the treated group with the control group, which has no treatment and no stress. This is not the best comparisons to do.

Also, a lot of the parameters that the authors measured, the differences between “B+S” and “P+S” groups are not significant. I do not believe the authors can safely arrive to their conclusion without carefully examining and explaining these non-significant caparisons data points.

The authors need to improve the introduction to get the reads prepared. For example, I do not have enough information about the parameters that the authors measured (superoxide, MDA, osmolytes, etc.). The authors should not assume that everyone who reads this paper has all the necessary background information as the authors do.

Similarly, the authors should do a better job writing down the details of the parameters they measured and why they are important for the traits that they wanted to monitor. For example, for the “3.7. Photobiology” part, it was very difficult for me to understand what exactly the authors did with the “ABS”, “ET” etc. measurement, and the Figure 5 was also not clearly labeled nor annotated. I went back to the materials and methods section, but I could only find some very brief descriptions about the equipment in section “2.9. Measurements of chlorophyll a fluorescence parameters”. I bet most of the readers would find it difficult to follow as well.

Some small tissues:

Format issue in Table 1, between Line 205-213.

Author Response

Authors’ Responses to Reviewer I Comments

The authors are greatly thankful to the learned reviewers for the insightful comments and suggestions for improving the scientific and technical quality of the manuscript. We hope that by addressing the reviewer’s comments, the manuscript meets the publication requirements of your esteemed journal. We have incorporated all the modifications suggested by the reviewers.

Reviewer Comments

Reviewer 1:

The authors treated cowpea seeds with either BABA or PEG as priming agents and measured parameters related to plant drought responses. The authors concluded that BABA primed seeds performed better than PEG primed seeds towards PEG stress.

 One big issue with the way the authors present their findings is that, they wanted to compare between BABA- and PEG-treated seeds, however, in all of their comparisons, they were trying to compare all the treated group with the control group, which has no treatment and no stress. This is not the best comparisons to do.

Response: We are grateful to the reviewer for providing valuable suggestions and critical feedback to improve our manuscript. We have tried our best to make changes in the manuscript as per the suggestions. We modified the comparisons between the treatments and included the percentage difference between BABA and PEG treatment under controlled and stressed conditions.

Also, a lot of the parameters that the authors measured, the differences between “B+S” and “P+S” groups are not significant. I do not believe the authors can safely arrive to their conclusion without carefully examining and explaining these non-significant caparisons data points.

Response: Even though there is no huge difference between BABA and PEG primed treatments, we can notice some significant changes in the activity of some important physiological parameters (ROS, MDA,Antioxidants) from which we can deduce the conclusion that BABA priming is better than PEG priming in cowpea seedlings subjected to PEG-induced osmotic stress.

 The percentage of increase in the generation of reactive oxygen species is less in BABA primed plants as compared to PEG primed plants and thereby there is a significant reduction in malondialdehyde content in BABA primed plants subjected to stress as compared to PEG primed plants.

Similarly, there is a significant increase in the content of glutathione, total phenolics, activity of superoxide dismutase (SOD) and catalase in BABA primed plants under stress condition. When expressed in percentage we found an overall improved performance in BABA primed plants subjected to PEG stress than PEG primed plants subjected to PEG stress. When are looking into the photosynthetic efficiency of plants, the electron transport flux per cross section (ETo/CSm) is higher in BABA primed plants under stress and energy dissipation flux per CS (DIo/CSm) were reduced due to the beneficial effect of BABA in cowpea.

The authors need to improve the introduction to get the reads prepared. For example, I do not have enough information about the parameters that the authors measured (superoxide, MDA, osmolytes, etc.). The authors should not assume that everyone who reads this paper has all the necessary background information as the authors do.

Response: Thank you for the suggestion. We modified the manuscript with some details of the physio-chemical parameters we considered. Hope it helps to understand the background information of the work.

Similarly, the authors should do a better job writing down the details of the parameters they measured and why they are important for the traits that they wanted to monitor. For example, for the “3.7. Photobiology” part, it was very difficult for me to understand what exactly the authors did with the “ABS”, “ET” etc. measurement, and the Figure 5 was also not clearly labeled nor annotated. I went back to the materials and methods section, but I could only find some very brief descriptions about the equipment in section “2.9. Measurements of chlorophyll a fluorescence parameters”. I bet most of the readers would find it difficult to follow as well.

Response: We are grateful to the reviewer for the valuable suggestions. We tried to improve the methodology part by providing the detailed protocol of the work done. Also, incorporated all the suggestions concerning fig. 5 including the details of equipment, parameters and their relevance.

Some small issues: Format issue in Table 1, between Line 205-213.

Response: As per your grateful suggestion we modified the table. Thank you for your valuable and critical comments, which has helped to improve the scientific and technical quality of our manuscript.

Thank you…

Reviewer 2 Report

Dear Colleagues

I have read with interest the manuscript of the article by K.P. Raj Aswathi et al. The authors have some new data on seed priming with PEG and BABA. However, I have some comments which the authors would like to consider.

1.         The methodological part is practically missing. There is no description of methods for determination of antioxidant enzyme activities and proline, sugars and other low-molecular-weight organic compounds.  Without a description of the methods of analysis, the experimental paper cannot be recommended for publication.

2.         In the classical sense, drought is the interaction of two factors lack of water and elevated temperature. There is, however, soil drought, which is a consequence of water stress. For this reason, I don't think it is correct to speak of drought in this paper. The authors are not studying the effects of drought, but osmotic stress in its purest form. This is what they should write about.

3.         Under osmotic stress, plants lose the ability, as under control conditions, to freely absorb water. This leads to the development of water deficiency in the cells.  When studying water stress, it is mandatory to evaluate the water status of the plant (it may be the tissue water content, RWC assessment, measurement of osmotic and/or water potential), as the primary target of water stress is the intracellular water. This was not done by the authors, which is a serious shortcoming of this study. PEG solutions should also be expressed in the osmotic potential of the solutions used, not in percentages of this substance.

4 The authors analyse the content of proline, soluble sugars and amino acids and put them in a separate section called Osmolytes, which is correct. Then comes the Non-enzymatic antioxidants section, where the authors put ascorbate, glutathione and phenols. At the same time, it is well known that all the osmolytes listed by the authors also exhibit antioxidant properties. For this reason, it would be more correct to consider all these compounds as non-enzymatic antioxidants, acting simultaneously as osmolytes, but it is not correct to separate the analyzed substances into osmolytes and non-enzymatic antioxidants.

5.         I cannot agree with the last sentence of the Introduction "In this work, our research team sheds light on the cross-talk between multiple signalling pathways and gives a clear cut understanding of the priming-induced cross-stress tolerance mechanism towards PEG-induced drought attributed through BABA", as this study did not investigate any signalling pathways, much less any interaction.

It is very difficult for me to reach a conclusion on this paper. I would recommend rejecting the manuscript as submitted, although I am not aware of the requirements of the journal Seeds, so I recommend a major revision of this paper.

 Kind regards

Author Response

Authors’ Responses to Reviewer II Comments

I have read with interest the manuscript of the article by K.P. Raj Aswathi et al. The authors have some new data on seed priming with PEG and BABA. However, I have some comments which the authors would like to consider.

We are grateful to the reviewer for providing valuable suggestions and critical feedback to improve our manuscript. We have tried our best to make changes in the manuscript as per your suggestions

The methodological part is practically missing. There is no description of methods for determination of antioxidant enzyme activities and proline, sugars and other low-molecular-weight organic compounds.  Without a description of the methods of analysis, the experimental paper cannot be recommended for publication.

Response: As per your valuable suggestion we have modified the methodology part and have written a detailed account about all the protocols.

In the classical sense, drought is the interaction of two factors lack of water and elevated temperature. There is, however, soil drought, which is a consequence of water stress. For this reason, I don't think it is correct to speak of drought in this paper. The authors are not studying the effects of drought, but osmotic stress in its purest form. This is what they should write about.

Response: We have addressed your comment. In this paper we focused on the effect of PEG-induced osmotic stress. PEG stress causes non availability of water, decreasing water absorption and causes desiccation, similar to actual drought stress. As a result, PEG is being frequently used in many studies as an artificial drought stress inducer.  Throughout the MS, we are describing about the osmotic stress caused by PEG.

As per your valuable suggestion we modified the title as well as the introduction part accordingly. In the revised manuscript we provide more emphasize to the effect of osmotic stress. 

Under osmotic stress, plants lose the ability, as under control conditions, to freely absorb water. This leads to the development of water deficiency in the cells.  When studying water stress, it is mandatory to evaluate the water status of the plant (it may be the tissue water content, RWC assessment, measurement of osmotic and/or water potential), as the primary target of water stress is the intracellular water. This was not done by the authors, which is a serious shortcoming of this study. PEG solutions should also be expressed in the osmotic potential of the solutions used, not in percentages of this substance.

Response: We have checked the tissue water status using vapour pressure osmometer and the data obtained is provided in the table (Table 1). Measurement of osmolality is a good parameter to assess the water potential of plants.

We also checked the osmotic potential of PEG-6000 solutions. 0%, 5%, 10%,15%,20%, 25% (w/v) corresponds respectively as 0 MPa, -0.2 Mpa, -0.45M Pa, -0.67 MPa, -0.9M pa and -1.12 MPa respectively.

The authors analyse the content of proline, soluble sugars and amino acids and put them in a separate section called Osmolytes, which is correct. Then comes the Non-enzymatic antioxidants section, where the authors put ascorbate, glutathione and phenols. At the same time, it is well known that all the osmolytes listed by the authors also exhibit antioxidant properties. For this reason, it would be more correct to consider all these compounds as non-enzymatic antioxidants, acting simultaneously as osmolytes, but it is not correct to separate the analyzed substances into osmolytes and non-enzymatic antioxidants.

Response: As per the suggestion of the reviewer we included osmolytes under non-enzymatic antioxidants.

I cannot agree with the last sentence of the Introduction "In this work, our research team sheds light on the cross-talk between multiple signalling pathways and gives a clear cut understanding of the priming-induced cross-stress tolerance mechanism towards PEG-induced drought attributed through BABA", as this study did not investigate any signalling pathways, much less any interaction.

Response: We modified the manuscript according to your suggestion.

It is very difficult for me to reach a conclusion on this paper. I would recommend rejecting the manuscript as submitted, although I am not aware of the requirements of the journal Seeds, so I recommend a major revision of this paper.

We are thankful to the learned reviewer for providing valuable suggestions and critical feedback. We have tried our best to make changes in the manuscript as per your suggestions.

Thank You…

Reviewer 3 Report

The paper concern an interesting issues of the influence of priming on the establishment of plant tolerance to stress factors. The paper presents interesting results. In my opinion, the manuscript is very well prepared. Disscusion is interesting for readers. It should also be appreciated that in manuscript the most recent publications are cited. A major remark concerns the language, some sentences are difficult to understand. I also have a few minor comments:
1) Were repetitions made for leaf osmolality, because there is no information about it? What value is given in table 1 - average of repetitions?
2) Although such information is in the methods, it can also be added in the figures caption what the error bars represent
3) The quality of Figure 5 can be improved, because it is not clear
4) Line 119: subscripts must be used

Author Response

Authors’ Responses to Reviewer III Comments

The paper concern an interesting issues of the influence of priming on the establishment of plant tolerance to stress factors. The paper presents interesting results. In my opinion, the manuscript is very well prepared. Disscusion is interesting for readers. It should also be appreciated that in manuscript the most recent publications are cited. A major remark concerns the language, some sentences are difficult to understand.

We are extremely thankful for the appreciation that you have mentioned about our work and manuscript preparation. We are greatly thankful to the learned reviewer for your precious time and valuable suggestion to improve the scientific and technical quality of our manuscript.

Response: Thanks for pointing out the need for language revision. We have edited the language throughout the manuscript with the help of Grammarly Editor Premium and hope this would suffice.

1) I also have a few minor comments: Were repetitions made for leaf osmolality, because there is no information about it? What value is given in table 1 - average of repetitions?

Thanks for pointing out the need for the information regarding the values provided in the table. The value is the average of three independent replicates. Also we have included the standard error of the values.

2) Although such information is in the methods, it can also be added in the figures caption what the error bars represent

Response: As per your valuable comments we have included the details in all the figure captions. Values represent the mean ± SE of three independent experiments. Alphabetical letters show significant difference between treatments performed using Duncan's multiple range test. 

3) The quality of Figure 5 can be improved, because it is not clear

Response: We have improved the figure quality of Figure 5.

4) Line 119: subscripts must be used

Response: Thanks for pointing out the typos. We modified it in the revised manuscript.

We greatly acknowledge you for your valuable comments and suggestion. We tried to incorporate all the suggestions and hope that, now the manuscript is improved and ready to be published in the prestigious journal.

Round 2

Reviewer 1 Report

The authors have addressed most of the questions and concerns that I had in the original manuscript.

Author Response

Response to the reviewer comments

Reviewer 1

The authors have addressed most of the questions and concerns that I had in the original manuscript.

Response: We are grateful to know that the manuscript is modified and hope that now it is suitable for publication.

Thank you…

Reviewer 2 Report

Dear Colleagues

I carefully read the answers of the authors and looked at the revised manuscript. The manuscript has been greatly improved. The authors took the recommendations seriously and took many of them into account. I cannot say that the manuscript has become perfect, but in the corrected form it can already be recommended for publication. At the same time, the centrifugation speed is usually expressed not in revolutions per minute, but in g. If the authors prefer to express the centrifugation speed in rpm, then it is necessary to indicate the brand of the centrifuge and its manufacturer in accordance with standard requirements.

With kind regards

Author Response

Response to the reviewer comments

Reviewer 2

I carefully read the answers of the authors and looked at the revised manuscript. The manuscript has been greatly improved. The authors took the recommendations seriously and took many of them into account. I cannot say that the manuscript has become perfect, but in the corrected form it can already be recommended for publication. At the same time, the centrifugation speed is usually expressed not in revolutions per minute, but in g. If the authors prefer to express the centrifugation speed in rpm, then it is necessary to indicate the brand of the centrifuge and its manufacturer in accordance with standard requirements.

Response: We are grateful to your suggestions. As per the suggestion we have expressed the centrifugation speed in g.

Thank you…
